# Isolation of Functional Human MCT Transporters in *Saccharomyces cerevisiae*

**DOI:** 10.3390/cells13181585

**Published:** 2024-09-20

**Authors:** Hajira Ahmed Hotiana, Karl Patric Nordlin, Kamil Gotfryd, Per Amstrup Pedersen, Pontus Gourdon

**Affiliations:** 1Department of Biomedical Sciences, Faculty of Health and Medical Sciences, University of Copenhagen, Maersk Tower 7-9, DK-2200 Copenhagen N, Denmark; 2Department of Biology, Faculty of Science, University of Copenhagen, Universitetsparken 13, DK-2100 Copenhagen OE, Denmark; 3Department of Experimental Medical Science, Faculty of Medicine, Lund University, Sölvegatan 19, SE-221 84 Lund, Sweden

**Keywords:** membrane proteins, overproduction, *Saccharomyces cerevisiae*, yeast, production platform, protein purification, human metabolism, human monocarboxylate transporters (hMCTs), solute carrier 16 (SLC16) family, aromatic amino acid transporters

## Abstract

Human monocarboxylate transporters (hMCTs) belong to the solute carrier 16 (SLC16) family of proteins and are responsible for the bi-directional transport of various metabolites, including monocarboxylates, hormones, and aromatic amino acids. Hence, the metabolic role of hMCTs is undisputable, as they are directly involved in providing nutrients for oxidation and gluconeogenesis as well as participate in circulation of iodothyronines. However, due to the difficulty in obtaining suitable amounts of stable hMCT samples, the structural information available for these transporters is limited, hindering the development of effective therapeutics. Here we provide a straightforward, cost-effective strategy for the overproduction of hMCTs using a whole-cell *Saccharomyces cerevisiae*-based system. Our results indicate that this platform is able to provide three hMCTs, i.e., hMCT1 and hMCT4 (monocarboxylate transporters), and hMCT10 (an aromatic amino acid transporter). hMCT1 and hMCT10 are recovered in the quantity and quality required for downstream structural and functional characterization. Overall, our findings demonstrate the suitability of this platform to deliver physiologically relevant membrane proteins for biophysical studies.

## 1. Introduction

Monocarboxylic acids such as L-lactate, pyruvate, and ketone bodies are major players in the cellular metabolism of all living cells [1,2]. The monocarboxylates are involved in fundamental energy homeostasis pathways, which entails the rapid movement of these compounds across cellular membranes. Depending on the tissue type, the influx and efflux across these membranes of monocarboxylates, especially lactic acid, must be tightly regulated [3,4]. Lactic acid is produced as a byproduct of glycolysis, and its transport across the cellular membrane is crucial for cellular metabolism and pH regulation.

Monocarboxylate transporters (MCTs) are members of the solute carrier 16 (SLC16) family of transporters and are responsible for the bi-directional transport of these essential metabolic solutes across the plasma membrane. Considering their broad transport specificity, MCT members play a pivotal role in maintaining energy homeostasis in all mammalian cells. The MCT family comprises 14 human proteins (Figure 1A). Key members include MCT1 and MCT4, which facilitate the bi-directional, passive transport of monocarboxylates (e.g., lactate, pyruvate and ketone bodies) in a proton-dependent manner [5]. MCT1 is expressed in a plethora of tissues, including the brain, heart, kidneys, liver, and skeletal muscles [1]. MCT1 is crucial in shuffling lactate to prevent a fall in cytosolic pH, thus contributing to the regulation of glycolytic energy production. In addition, along the gastrointestinal track, MCT1 is expressed at varying levels in the epithelial cells on both basolateral and apical membranes, where it facilitates the uptake of acetate, propionate, and butyrate. Conversely, MCT4 displays a more cell- and tissue-restricted expression profile, and is predominantly expressed in tissues that rely on high levels of glycolysis to meet the energy requirements of cells. Two other important and metabolically relevant MCTs comprise MCT10, which mediates the proton-independent transport of the aromatic amino acids tyrosine (Tyr), tryptophan (Trp), and phenylalanine (Phe) across cellular membranes, and MCT8, which is a proton-independent T3 and T4 thyroid hormone transporter [6,7].

The malfunction of MCTs is implicated in the development of a wide spectrum of pathophysiological conditions, including several metabolic diseases. The failed silencing of MCT1 in pancreatic β-cells was associated with exercise-induced hyperinsulinism (EIHI), hypoglycemia, and active ulcerative colitis [10,11,12]. Moreover, the altered expression of MCT1 and 4 has been linked to mechanisms of weight gain, and MCT1 deficiency was found to manifest with recurrent ketoacidosis [11,13,14]. Finally, incorrect MCT1–ancillary protein complex formation may potentially contribute to the progress of subnormal conditions, as suggested for MCT1–basigin interaction in the development of fatigue [15]. Functional defects of MCT10, disrupting its role in the maintenance of aromatic amino acid homeostasis, have been proposed to contribute to the development of blue diaper syndrome caused by insufficient Trp absorption in the kidneys [16].

Although MCTs represent attractive drug targets, only a limited number of SLC16-targeting compounds are currently available due to the sparse structural information available for MCTs. Thus, it is of imminent importance that the architectural details of SLC16 members are deciphered not only to understand their molecular mechanism of action but also to permit fine-tuned, structure-based drug design.

Membrane proteins are notoriously difficult to produce in heterologous expression systems and, as such, the progress of structural studies of MCTs has been limited, and models generated by AlphaFold or similar protein structure prediction software do not produce models of sufficient accuracy. MCTs adopt a conformation characteristic of the major facilitator superfamily (MFS), with 12 transmembrane helices (TMs) and intracellularly located N- and C-termini. The 12 TMs are arranged into two bundles with six membrane spanning segments, with a pseudo-two-fold symmetry linked via a long intracellular loop connecting TMs 6 and 7. Additionally, to guide expression and trafficking, MCT1 and MCT4 can associate with ancillary glycoproteins and, depending on the tissue, form a complex with either basigin or embigin [5,17]. However, this assembly does not seem to be a perquisite for the transport function [18]. To date, MCT10 is not known to associate with any ancillary protein [2].

The experimentally obtained structural information for MCTs is limited to a bacterial L-lactate transporter from *Syntrophobacter furmaroxidans* (*Sf*MCT), a cryo-EM structure of human MCT2, and structures of the human MCT1–basigin complex [8,19,20]. All structures support an overall fold characteristic of the MFS family as predicted with 12 TMs. However, the MCT1–basigin cryo-EM structure suggests a 1:1 heterodimer assembly, while the inward-open MCT2 structure revealed a homodimeric organization [8,20]. Nonetheless, MCT1 and MCT2 can be superimposed with an RMSD of 1.1 Å over the aligned C_α_ atoms, indicative of only minor structural differences between the members of this MCT subclass. On the other hand, structural comparison between human MCTs and SfMCT reveals major differences. Despite a similar location of the cargo-binding pocket, residues that form the binding pocket are not conserved. In addition, while SfMCT has two cargo-binding sites, hMCT1 only has one, as the other binding site is occupied by Lys38 in hMCT1.

*Escherichia coli* has traditionally been the system of choice to express and produce membrane proteins for downstream studies due to its easy genetic manipulation, fast growth rate, and comparatively low costs [21,22,23]. However, in terms of producing recombinant proteins from higher organisms, *E. coli*-based platforms display major shortcomings and, often, expression systems derived from a eukaryotic origin such as yeast, insect, or mammalian-based expression platforms are required [24,25]. Nevertheless, in contrast to *E. coli* systems, the establishment of insect and mammalian cell-derived expression systems is expensive, laborious, and time consuming. In this scenario, yeast systems provide a cheap platform that is easy to set up and use, with the inherent ability of the large-scale production of properly folded eukaryotic proteins with post-translational modifications that are more alike to those of higher organisms [21,22]. Hence, yeast represents an attractive host for the large-scale production of human membrane protein targets, allowing downstream structural, functional, and drug-discovery studies for many physiologically relevant proteins including MCTs [26,27,28,29].

Here, we report the development of a straightforward, economical, and rapid procedure employing a *Saccharomyces cerevisiae*-based platform for the overproduction of human MCTs. We attempted to achieve the expression of three selected hMCTs that play an imminent role in human metabolism: hMCT1, 4, and 10. Expression constructs using yeast-optimized codons were designed with a C-terminal His_10_ tag along with a tobacco etch virus (TEV) protease cleavage site and a green fluorescent protein (GFP) reporter. Using GFP fluorescence as a marker to assess the expression levels, solubilization efficiency, and protein stability, we initially set out to identify the selected MCTs for large-scale protein production. The most promising targets, i.e., MCT1 and MCT10, were purified using affinity chromatography, and the sample homogeneity was evaluated using size-exclusion chromatography. Further, we investigated the effects of relevant solutes on the expression and stability of protein samples. Taken together, our results demonstrate that *S. cerevisiae* is a suitable host for the overproduction of functionally active hMCTs for downstream structural, functional, and drug-discovery studies.

## 2. Materials and Methods

### 2.1. Cloning and Engineering of Expression Plasmids

cDNA encoding full-length hMCTs was codon optimized for *S. cerevisiae*-based expression using the OptimumGene^TM^ algorithm and purchased from GenScript (Piscataway, NJ, USA). Codon optimization takes into account a variety of parameters related to transcription, translation, and protein folding. cDNAs along the GFP fragment were PCR-amplified using AccuPol DNA polymerase (Amplicon, Odense, Denmark). hMCTs and GFP fusion expression constructs were generated employing homologous recombination by co-transforming the corresponding hMCT/GFP PCR-amplified fragments and a *Bam*H1-, *Hin*dIII-, and *Sal*I-digested pEMBLyex4 vector directly in the *S. cerevisiae* strain PAP1500 (Figure 2A,B) [30]. All engineered hMCT constructs encoded a C-terminally fused tag containing tobacco etch virus (TEV) protease cleavage site (ENLYFQ↓SQF), green fluorescent protein (GFP), and a decahistidine (His_10_) sequence for downstream affinity purification (Figure 2C). Transformants were selected on agar plates containing minimal medium supplemented with leucine (60 mg/L) and lysine (30 mg/L). The integrity of the transformants was confirmed by DNA sequencing on isolated plasmids.

### 2.2. Small-Scale Overproduction of hMCTs and Live-Cell Imaging

hMCT–TEV–GFP–His_10_ constructs were produced in the *S. cerevisiae* expression strain PAP1500 as previously reported [27,29]. For small-scale screening, hMCT1–TEV–GFP–His_10_, hMCT4–TEV–GFP–His_10_, hMCT8–TEV–GFP–His_10_, and hMCT10–TEV–GFP–His_10_ were expressed and grown in a 2 L cell culture using shaker flasks. Briefly, for each construct, a single colony of transformant was used to inoculate 5 mL of SD medium containing glucose (2 g/L), leucine (60 mg/L), and lysine (30 mg/L) and grown for 16 h at 30 °C until the OD_450 nm_ reached approximately 0.5–1.0. Next, 0.2 mL of the culture was transferred to 5 mL minimal medium without leucine and grown at 30 °C for 24 h to increase the plasmid numbers. Subsequently, the culture was scaled up to 100 mL in the same medium, grown for an additional 24 h, and used to inoculate 2 L of medium supplemented with amino acids (alanine (20 mg/L), arginine (20 mg/L), aspartic acid (100 mg/L), cysteine (20 mg/L), glutamic acid (100 mg/L), histidine (20 mg/L), lysine (30 mg/L), methionine (20 mg/L), phenylalanine (50 mg/L), proline (20 mg/L), serine (375 mg/L), threonine (200 mg/L), tryptophan (20 mg/L), tyrosine (30 mg/L), valine (150 mg/L), glucose (10 g/L), and glycerol (3%, *v*/*v*). Following consumption of glucose, the temperature was lowered to 15 °C, and protein expression was induced by supplementing the culture with 2% (*w*/*v*) (final concentration) galactose and allowed to grow for another 48–72 h. Subsequently, yeast cells were harvested by centrifugation at 5500× *g* for 15 min at 4 °C. Cell pellets were resuspended in 0.9% (*w*/*v*) NaCl before final harvesting at 3000× *g* for 10 min at 4 °C and stored at −80 °C. For small-scale screenings, a 2 L yeast culture typically yielded 6–8 g of wet cell pellet. Localization of expressed hMCT–TEV–GFP–His_10_ was performed by live-cell bioimaging of GFP fluorescence in vivo using Nikon Eclipse E600 microscope (Nikon, Tokyo, Japan) equipped with an Optronics camera, (Muskogee, OK, USA).

### 2.3. Preparation of S. cerevisiae Crude Membranes and In-Gel Fluorescence

For small-scale screening of hMCT–TEV–GFP–His_10_ expression and purification, thawed yeast cell pellets were resuspended in ice-cold solubilization buffer (SB: 20 mM Tris-HCl pH = 7.0, 200 mM NaCl, 20% (*v*/*v*) glycerol, 5 mM β-mercaptoethanol (BME), 1 mM phenylmethylsulfonyl fluoride (PMSF), and SigmaFASTTM protease inhibitor cocktail (Sigma-Aldrich, St. Louis, MO, USA). Resuspended cell pellets were mixed with glass beads (diameter 0.4–0.8 mm) and subjected to rigorous shaking for 8 rounds of 2 min each on a LGG uniTEXER vortex. Following cell disruption, using vacuum filtration, the supernatant was collected, and the glass beads were washed in ice-cold solubilization buffer. Unbroken cells and cell debris were separated using centrifugation at 2000× *g* for 10 min at 4 °C. Crude membranes were then pelleted via ultra-centrifugation at 230,000× *g* for 4 h at 4 °C, resuspended in solubilization buffer using a Potter-Elvehjem homogenizer, and stored at −80 °C until purification. In-gel fluorescence for the GFP-tagged proteins was detected on crude membrane samples resolved by 4–20% SDS-PAGE (Thermo Fisher Scientific, Waltham, USA) where the gel was immediately visualized using the ImageQuant LAS 4000 imaging system at 400 nm wavelength.

### 2.4. Detergent Screening and F-SEC Analysis

Isolated membranes from all the hMCT–TEV–GFP–His_10_ constructs were subject to detergent screening to test most efficient solubilization strategy as previously described [27]. Briefly, the total protein concentration was estimated using the Bradford Assay (Sigma-Aldrich) and the crude membranes were diluted in solubilization buffer to a final concentration of 1 mg/mL. Small-scale detergent screens were performed using 0.5 mL of crude membranes rigorously rotated (120 rpm) for 90 min at 4 °C with 1 % (*w*/*v*) final concentration of n-hexadecyl-phosphocholine (FC-16), n-dodecyl-D-maltoside (DDM), 5-Cyclohexyl-1-Pentyl-β-D-maltoside (Cymal-5), n-nonyl-β-D-glucopyranoside (NG), n-octyl-β-D-glucopyranoside (OG) (from Anatrace, Maumee, USA), and 4-trans-(4-trans-propylcyclohexyl-cyclohexyl a-maltoside (PCC) (Glycon Biochemicals, Germany) with or without 0.2% (*w*/*v*) of cholesteryl hemisuccinate Tris salt (CHS) dissolved in solubilization buffer were used for the screening. Subsequently, insoluble material was removed by ultracentrifugation at 50,000× *g* for 20 min at 4 °C. A total of 20 µL of supernatant collected from each protein was used to directly measure GFP fluorescence (excitation 485 nm, emission 520 nm) to assess solubilization efficiency. Solubilized supernatant was resolved on 4–20% SDS-PAGE and visualized on ImageQuant LAS 4000 imaging system to determine the molecular weight of the GFP fluorescent protein.

Fluorescence-detection size-exclusion chromatography (F-SEC) was carried out to determine dispersity of the sample solubilized from membranes in different detergents as previously reported. Briefly, 150 µL of solubilized membranes was filter-purified by centrifugation at 30,000× *g* for 10 min. Post-filtration sample was analyzed using size-exclusion chromatography (SEC), applying a Superose 6 10/300 GL column (Cytiva, Marlborough, USA) equilibrated with 50 mM Tris-HCl pH = 8.0, 300 mM NaCl, 10% (*v*/*v*) glycerol, 2 mM BME, and 0.03% (*w*/*v*) DDM attached to ÄKTA Pure system (Cytiva) equipped with a RF-20 A fluorescence detector (Shimadzu, Kyoto, Japan).

### 2.5. Large-Scale Production of hMCT10

Large-scale production of hMCT10–TEV–GFP–His_10_ was carried out in 15 L bioreactors as previously published [26]. Briefly, 100 mL of the yeast culture, as described for small-scale expression, was scaled up to 1 L in the same medium and grown for 16 h at 30 °C. Subsequently, the culture was used to inoculate 10 L of similar medium supplemented with 3% (*w*/*v*) glucose, 3% (*v*/*v*) glycerol, additional amino acids (except leucine), and inorganic salts and vitamins, and grown in Applikon fermenters connected to an ADI 1030 Bio Controller (Applikon Biotechnology, Delft, Netherlands), with the culture pH automatically maintained at 6.0. Subsequently, 18 h after inoculation, the cultures were supplemented with 1 L of 20% (*v*/*v*) glucose to further boost cell growth. Upon depletion of glucose, protein expression was induced by adding galactose to a final concentration of 2% (*w*/*v*). Cells were allowed to grow for another 96 h at 15 °C before harvesting through centrifugation at 3000× *g* for 10 min and stored at −80 °C until further use. A typical 10 L yeast culture yielded approximately 200 g of wet cell pellet.

### 2.6. Large-Scale Purification of hMCT10

Large-scale purification of hMCT10–TEV–GFP–His_10_ was carried out using immobilized metal ion affinity chromatography (IMAC) based on crude membranes isolated from 50 g of fermenter-grown yeast cells. Membranes were solubilized for 4 h at 4 °C using a final concentration of 1% (*w*/*v*) t-PCCam and 0.2% (*w*/*v*) CHS dissolved in SB buffer in the presence of 5 mM Phe. The unsolubilized fraction was removed via ultracentrifugation at 118,000× *g* for 1 h at 4 °C. The solubilized membranes for hMCT10–TEV–GFP–His_10_ were then diluted 2 times in SB buffer to reduce the detergent concentration, and the NaCl concentration was adjusted to 500 mM to diminish unspecific binding to IMAC column. The prepared sample was loaded onto a 5 mL HisTrap HP column pre-equilibrated with IMAC buffer A (25 mM Tris-HCl pH = 8.0, 300 mM NaCl, 10% (*v*/*v*) glycerol, 2 mM BME, 0.01% PCC, and 2 mM Phe) attached to an Äkta Pure system (both from GE Healthcare). Bound protein was eluted in IMAC buffer B (25 mM Tris-HCl pH = 8.0, 300 mM NaCl, 10% glycerol, 2 mM BME, 0.01% (*w*/*v*) PCC, 2 mM Phe, 500 mM imidazole) by applying a linear gradient of imidazole (50–500 mM). The peak fractions from the elution were pooled and the protein concentration was estimated. Typically, 50 g of fermenter cells yielded about 20 mg of IMAC pure hMCT10–TEV–GFP–His_10_ protein. Subsequently, the GFP–His_10_ tag was cleaved trough 16 h incubation with TEV–His_10_-tagged protease mixed with the protein sample at a ratio of 1:5 (*w*/*w*) in a dialysis tube (Thermo Scientific, Waltham, USA) with dialysis against IMAC buffer A supplemented with 20 mM imidazole. Following cleavage, reverse IMAC (RIMAC) was performed to separate the cleaved protein (hMCT10) from the uncleaved hMCT10–TEV–GFP–His_10_, the free GFP–His_10_ tag, and the TEV–His_10_ tagged protease. Briefly, the salt concentration in the cleaved sample was adjusted to 300 mM and the imidazole concentration to 50 mM before loading the sample onto pre-equilibrated (IMAC buffer A) 5 mL HisTrap HP columns (Cytiva, Marlborough, USA). Next, the flow-through was collected and concentrated using Vivaspin concentrators (MWCO 50 kDa; Sartorius, Gottingen, Germany) to ~8 mg/mL. Subsequently, as a final stage of purification, concentrated RIMAC pure protein was applied to equilibrated (SEC buffer: 50 mM Tris-HCl, 200 mM NaCl, 5% (*v*/*v*) Glycerol, 2 mM BME, 0.0025% (*w*/*v*) LMNG) Superdex 200 Increase 10/300 column (Cytiva, Marlborough, USA). Individual SEC fractions were resolved on a 4–20% Tris-Glycine (Thermo Fisher, Waltham, USA) SDS-PAGE, and the protein purity was estimated.

## 3. Results

### 3.1. Overproduction of hMCT-TEV-GFP-His Fusions in a S. cerevisiae-Based System

Three key members of the MCT family, i.e., hMCT1, hMCT4, and hMCT10, are widely recognized as critical players in fundamental cellular metabolic pathways such as glycolysis, gluconeogenesis, and fatty acid homeostasis [14,17]. Despite a long-standing interest in understanding the molecular mechanism of these transporters, a lack of cost-effective and efficient strategies to express and purify these proteins has hindered progress. Therefore, MCTs have proven to be difficult targets for structural and functional studies [31].

As the identification of a suitable cost-effective expression platform is often a bottleneck for downstream structural studies, we set out to determine if our established *S. cerevisiae*-based system could effectively produce hMCTs in a functional form. Expression constructs were designed to encode the full-length sequences of hMCT1, hMCT4, and hMCT10 codon-optimized for *S. cerevisiae* and fused to a cleavable C-terminal TEV–GFP–His_10_ tag, allowing downstream localization, quantification, solubilization screening, and affinity purification. The recombinant plasmid was engineered to exploit the CG-P hybrid promoter on the pEMBLyex4 vector, enhanced by the Gal4 p transcriptional activator overexpressed in the PAP1500 yeast strain. Additionally, selection for leucine autotrophy was used to boost the efficiency of the system by proliferating the plasmid numbers before the expression induction.

High protein yields are crucial for many downstream structural and functional studies and, therefore, in the initial step, it is critical to assess the production efficiency of a heterologous system. To this end, we initially attempted 48 h induction in 2 L cultures grown in flasks. Since the constructs were fused to a GFP tag, we were able to follow the generated fluorescence to easily quantify and localize the amount of protein produced by the system. The four constructs were expressed in the cells with highly promising yields ranging from 10 to 16 mg of protein per liter of cell culture (Figure 2D,E). Additionally, we were able to localize the expressed protein in different compartments of the cell. Live-cell bioimaging fluorescent micrographs of the cells indicated that hMCT1 and hMCT4 accumulation was present in both the plasma membrane and the intracellular compartments, while hMCT10 was generally localized to the intracellular compartments only. This variation in compartmentalization arises from the fact that these transporters are present in different cell membranes and target-specific compartmentalization is often observed for recombinant proteins produced in *S. cerevisiae*. Overall, for all three targets, the level of expression was highly promising for downstream biophysical and structural efforts.

### 3.2. Detergent Screening and F-SEC Analysis

Membrane protein purification heavily relies on the ability to extract the targets from their native lipid environment while maintaining the stability of the proteins in solution. A variety of surfactants are widely used to perform solubilization and purification, depending on the expression system, protein origin, and downstream application [32]. Initially, to identify the most suitable solubilization strategy, we employed detergents belonging to two classes, non-ionic and zwitterionic (Figure 3). All detergents were tested at a final concentration of 1% (*w*/*v*) and supplemented with 0.2% (*w*/*v*) CHS. The solubilization efficacies for the targets indicated that the zwitterionic detergent FC-16 was able to extract the proteins with near 100% efficiency. The non-ionic detergents DDM and PCC provided a highly suitable level of membrane extraction for hMCT1 (greater than 75%). Under identical detergent concentrations, PCC but not DDM recovered high levels of hMCT10 (~80%). The tested non-ionic detergents were the least successful in solubilizing hMCT4, extracting ~20% and ~30% in DDM and PCC, respectively.

A suitable choice of detergent also depends on its ability to maintain the target protein in a stable and homogenous state. To characterize the effect of the detergents on the protein and their usefulness for downstream biophysical and structural studies, we again relied on the TEV–GFP–His_10_ fusion tag and carried out an in-depth fluorescence-detection size-exclusion chromatography (F-SEC) analysis in the presence of 0.03% (*w*/*v*) DDM in the buffer to minimize sample aggregation. The F-SEC profiles indicated that the targets displayed sharp, symmetrical peaks with almost no aggregation when solubilized in the zwitterionic detergent FC-16, suggesting monodispersity of the hMCTs (Figure 3B–D). However, this detergent is considered harsh and is typically not suitable for downstream studies of membrane proteins. Two of the target proteins, hMCT1 and hMCT10, demonstrated promising F-SEC profiles in both DDM and PCC with almost no aggregation, and a single, symmetrical peak corresponding to a stable protein sample. hMCT4 also showed similar F-SEC profiles for the two detergents, although the peaks were much broader compared to hMCT1 and hMCT10. A large void peak was also present for hMCT4 (indicated in Figure 3C at ~8 mL), suggesting substantial aggregation of the protein sample. Moreover, apart from the void peak, DDM- and PCC-solubilized hMCT4 was eluted in two major fluorescence peaks, indicating the presence of higher oligomer (possibly dimer) and lower oligomer (possibly monomer) states. Overall, based on this initial analysis, the solubilization efficiency and monodispersity of the solubilized samples highlighted hMCT1 and hMCT10/MCT10 as the most promising targets for downstream studies.

### 3.3. Solubilization and Stability Screening in the Presence of Ligands

As previously mentioned, hMCTs are widely known to be involved in the transport of monocarboxylates (hMCT1/hMCT4) and aromatic amino acids (hMCT10). In this light, we set out to test if the presence of solutes during solubilization and F-SEC analysis can impact the efficiency of extraction and overall protein stability. hMCT1 and hMCT4 are primarily known to carry out the transport of monocarboxylates, including pyruvate and lactate. Hence, the solubilization efficacies of hMCT1 and MCT4 were tested in the presence of DDM and the related non-ionic detergent 5-cyclohexyl-1-pentyl-β-D-maltoside (Cymal-5) (Figure 4). The results showed that pyruvate but not lactate was able to enhance the solubilization efficiency of hMCT1 in both DDM and Cymal-5. On the contrary, for hMCT4, the presence of either solute resulted in a considerable two- and eight-fold increase in the cases of DDM and Cymal-5, respectively.

The F-SEC analysis of hMCT1 solubilized in DDM indicated that the increased solubilization in the presence of pyruvate did not correlate with enhanced protein stability (Figure 4B,C). hMCT1 in the presence of no solute was eluted with a void peak and a broad peak likely corresponding to a dimer. Even though lactate was less potent in solubilizing hMCT1 than pyruvate, the F-SEC chromatogram indicated a more homogenous sample with no aggregation. Protein solubilized in the presence of pyruvate, on the other hand, displayed a significant amount of aggregation and the main peak was shifted towards a later elution volume, indicating the presence of a lower oligomeric state (monomer). F-SEC profiles of hMCT1 solubilized in Cymal-5 in the presence of solutes also indicated that, even though the presence of solutes increased the solubility of the protein, they also lead to aggregation of the protein sample and a shift towards a lower-molecular-weight oligomer (monomer).

In the case of hMCT4 solubilized in DDM, no major change was observed in the F-SEC profiles in the presence of either solute. However, for hMCT4 solubilized in Cymal-5, the addition of lactate reduced both the aggregation and population of higher oligomers in the sample.

hMCT10 is primarily responsible for the transport of aromatic amino acids, although it is also known to have affinity for some monocarboxylates. We tested the impact of the presence of the aromatic amino acids Phe and Trp on the solubilization and protein homogeneity of hMCT10 (Figure 5A). Two different subclasses of non-ionic detergents were included in these tests, DDM and PCC, as well as the glucosides NG and OG. For hMCT10 solubilized in DDM and PCC, the addition of aromatic amino acids resulted in a boost in the efficacy towards 100%. A similar pattern was observed for hMCT10 extracted in NG. Surprisingly, hMCT10 extracted in OG in the presence of Phe exhibited a higher solubility efficacy, whereas the presence of Trp resulted in a significant loss in solubilization efficacy (~40%).

The F-SEC profiles of hMCT10 solubilized in DDM indicated no effect of the presence or absence of Phe on the stability of the protein. However, in the case of PCC-solubilized hMCT10, the presence of Phe smoothed the F-SEC profile, resulting in a decrease in the aggregated protein fraction and enriching the monomeric state (Figure 5C). In the case of tryptophan, for both the non-ionic detergents, the F-SEC profiles indicated a complete loss of the aggregate fraction and monodispersity was observed for the solubilized hMCT10. In the case of the glucosides, the presence of tryptophan did not have a major impact on the homogeneity of the protein.

Keeping in light the overall findings from the solubilization screens and F-SEC analysis, we decided to aim for large-scale purification of the aromatic amino acid transporter hMCT10–TEV–GFP–His_10_ using PCC as a detergent in the presence of the aromatic amino acid phenylalanine.

### 3.4. Purification of hMCT10-TEV-GFP-His

The large-scale purification of hMCT10–TEV–GFP–His_10_ was performed using affinity chromatography on crude membranes isolated from 40 g of fermenter-grown *S. cerevisiae* cells. Isolated membranes were solubilized in a final concentration of 1% (*w*/*v*) PCC + 0.2% (*w*/*v*) CHS in the presence of 2 mM Phe (4 h at 4 °C). hMCT10–TEV–GFP–His_10_ was eluted at a concentration of 300 mM imidazole with an overall yield of almost 10 mg following immobilized metal ion affinity chromatography (IMAC), although some protein was lost in the flow-through during the sample application (Figure 6). Analysis revealed a rather pure sample that was present primarily in a monomeric form following SDS-PAGE (Figure 6B). In the top two fractions, however, higher oligomeric states of the protein were also visible. Subsequently, the protein was subjected to treatment with TEV protease and reverse IMAC (RIMAC) purification to eliminate the remaining impurities and to remove the TEV–GFP–His_10_ tag, re-exposing the sample to the affinity resin and collecting the flow-through. The RIMAC chromatogram indicated successful cleavage of the C-terminal tag, as all of the hMCT10 protein was eluted during sample application. Additionally, SDS-PAGE comparison analysis of the before-cleavage (BC) and the after-cleavage (AC) samples indicated a shift in the overall size of the protein (from 85 to 55 kDa), further indicating a successful cleavage (Figure 6A–D).

The robustness of this purification procedure was assessed through the utilization of DDM for solubilization, and then exchanging of the detergent for lauryl maltose neopentyl glycol (LMNG) during the IMAC procedure, as this detergent is frequently applied for the single-particle analysis of membrane proteins. Analysis of the homogeneity and stability of hMCT10 through SEC shows that the protein elutes as a single well-defined peak in the presence of LMNG (Figure 6E). Further analysis of the collected SEC fractions via SDS-PAGE shows a relatively pure hMCT10 sample (Figure 6F). Interestingly, higher oligomers of hMCT10 can be seen in the SDS-PAGE analysis, suggesting that hMCT10 may not operate as a monomer.

## 4. Discussion

Membrane proteins constitute about 30% of the total human proteome and are known to be critically involved in a variety of cellular processes [28,29]. Hence, there is an ever-increasing interest in studying their structural properties to understand their function and their potential uses for drug design. Despite this deep-rooted interest, producing membrane proteins, especially those of human origin, in sufficient quantity and quality presents a bottleneck for structure–function drug-discovery studies. Therefore, compared to their soluble counterparts, membrane proteins remain relatively poorly characterized, conveying the need for the regular re-development of efficient, easy to handle, cost-effective, and reproducible means to produce significant yields of recombinant human membrane proteins.

Herein, we report a strategy to produce sufficient yields of the challenging hMCTs for downstream biophysical efforts. Despite decades of interest and research, the structural information available for hMCTs is limited and, for years, researchers depended on homology models to design small molecular inhibitors against this crucial class of protein. In the past few years, however, some experimental structures were determined which provided in-depth architectural information. These include structures of a distant bacterial homolog SfMCT, the pyruvate transporter hMCT2, and the monocarboxylate transporter hMCT1. While these structures represented a crucial breakthrough, they also highlighted the need to explore alternative, fast, and economical means of producing clinically relevant hMCTs for future structural and functional studies.

The hMCT2 and hMCT1 structures were determined using proteins expressed in mammalian expression system and insect Sf9 cells, respectively. While both these hosts are highly efficient in producing human proteins, they are also expensive and generate limited amounts of protein. The bacterial homolog SfMCT, on the other hand, was produced in *E. coli*, which is both a cost-effective and easy to set up platform with which to produce proteins. However, its ability to produce stable and functional recombinant human proteins is limited for eukaryotic targets. In the present study, we aim to bridge the gap between economical and easy to set up protein production platforms to recover sufficient quantities of functionally relevant hMCTs. We show that the *S. cerevisiae*-based heterologous expression system offers high yields of hMCTs while requiring simpler culture conditions and significantly lower costs. Previously, other classes of membrane proteins have been successfully produced in this platform and used to deliver crystal (human aquaporin 10) and cryo-EM (human chloride channel ClC-1) structures [26,28].

We adopted a systematic approach to gain the efficient overproduction of hMCTs in the *S. cerevisiae* expression system, starting with optimal plasmid construction, yield investigation, detergent screening, and protein stability optimization. Downstream of overproduction, we applied affinity purification, SEC-based assessment, and, ultimately, functional analysis of the most promising target.

All hMCT constructs were engineered with full-length sequences codon-optimized for *S. cerevisiae*, a strategy commonly used to boost production levels. Additionally, a C-terminal-cleavable TEV–GFP–His_10_ tag was included in all constructs to allow downstream yield estimation, protein localization, detergent screening, solubility assessment, and, ultimately, affinity purification. The initial small-scale expression screen carried out in flasks immediately indicated the suitability of the system to express the selected hMCTs with production levels ranging from 7.5 to 16 mg of protein per liter of cell culture. Such yields are highly encouraging for the recombinant production of human membrane proteins. We were also able to observe differences in the localization of the four hMCTs, with hMCT1 and hMCT4 localizing in both the plasma membrane and the intercellular compartments while hMCT10 localized primarily in the plasma membrane. The accumulation of recombinant human membrane proteins in different compartments is common in such a system and is in accordance with the previously reported cellular locations of hMCTs.

In order to achieve a sufficient quantity and stability of hMCTs, we sought to identify suitable membrane-extraction strategies using a commonly used classes of detergents. We observed that, overall, hMCTs are not highly resistant to extraction from yeast membranes, as the efficiency of extraction was always over 20%, even under the least-suitable conditions. The zwitterionic detergent FC-16 provided the highest solubility of almost 100% for all four hMCTs tested with no additional optimization. However, this detergent is known to be harsh for membrane proteins and, therefore, was not considered for downstream purification studies. Initially we observed that the presence of cholesterol (CHS) during solubilization had a positive impact on the efficiency of solubilization and protein stability for all four targets. This suggests that CHS enhances the fluidity of the yeast membranes, allowing more efficient extraction, and it may even associate and stabilize the hMCTs.

As hMCTs are solute transporters, we predicted that the presence of suitable solutes during detergent and stability screens may improve solubilization efficiency. We observed that, in the case of the monocarboxylate transporters hMCT1 and hMCT4, the addition of lactate and pyruvate at a concentration of 10 times the known K_d_ resulted in a significant boost in the membrane extraction of each of the two targets. However, this enhancement was not always accompanied by an improvement in the protein stability as visualized in the F-SEC analysis, indicating that hMCT1 and hMCT4 may require additional binding partners to attain stability. Indeed, hMCT1 and hMCT4 are known to have as ancillary proteins embigin and basigin, which are required to achieve stability and functionality. Thus, additional optimization of the screen with the ancillary proteins either co-expressed or included during solubilization could be carried out to even further augment the protein stability. However, for the purpose of our studies, the improved solubilization and stability achieved in the presence of lactate and pyruvate alone was sufficient. Similarly, the presence of the aromatic amino acids Phe and Trp greatly stimulated the solubilization efficiency of hMCT10 from yeast membranes. Interestingly, especially in the case of Trp, this boost in membrane extraction was accompanied by an increase in protein stability, an observation that is congruent with our earlier hypothesis, as hMCT10 is not known to have any ancillary proteins. However, since Trp is known to have an affinity for nickel and, hence, may interfere during affinity chromatography, we resorted to using Phe throughout solubilization and purification of hMCT10 [33,34].

Affinity chromatography of the clinically relevant hMCT10 yielded significant yields of reasonably pure C-terminal-tagged protein. The placement of the tag on the N- versus the C-terminus was shown to improve protein quantities in previous studies and can also be an asset in cases where the TEV cleavage is inefficient. However, in the case of hMCT10, the TEV cleavage was almost complete with the designed construct, hence eliminating the need to redesign the construct. Nevertheless, additional studies can be carried out with an N-terminal tag to assess its effect on protein yields. Moreover, a quality of the cleaved hMCT10 was achieved after the RIMAC was assessed in the presence of PCC + CHS and Phe. The cleaved hMCT10 in the presence of Phe displayed a highly encouraging homogenous and symmetrical SEC peak complemented by high purity, as visualized using SDS-PAGE. Finally, at least for hMVT10, this procedure can also be applied using DDM, and the protein is also compatible with detergent exchange to LMNG.

Collectively, our results provide the basis to set up an economical, efficient, and easy-to-handle *S. cerevisiae*-based platform to produce milligrams of functionally active recombinantly produced hMCTs in the purity and quality necessary for structural and functional studies. Thus, our strategy provides a reasonable alternative to current methodologies of producing hMCTs for downstream biophysical characterization efforts.

## Figures and Tables

**Figure 1 cells-13-01585-f001:**
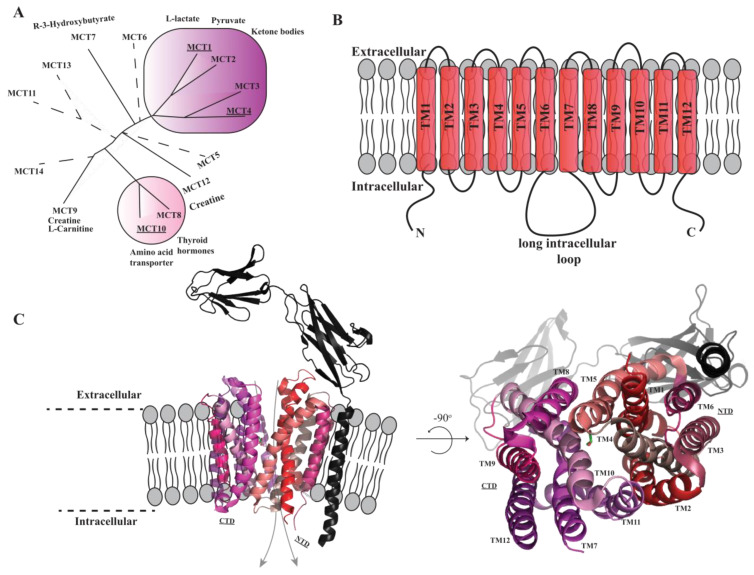
Overview of the human monocarboxylate transporter (hMCT) family. (**A**) Phylogenetic distribution of all 14 members of the human monocarboxylate transporter family and associated solutes. Well-characterized members of the family are emphasized using circles. The members of the family included in this study are underlined. (**B**) Topology of hMCT transporters with 12 transmembrane helices, a long intercellular loop connecting TM6 and TM7, and intercellular N- and C-termini. (**C**) Overall structure of the outward-open hMCT1 transporter with the N- and C-termini enclosing a cavity that is exposed at the extracellular side (outward-open), solved in complex with its ancillary protein basigin (shown in black) bound to lactate (PDB: 6 LZ0) [8]. Two separate views are shown on the left and right. Figures of hMCT1 were made using PyMOL [9].

**Figure 2 cells-13-01585-f002:**
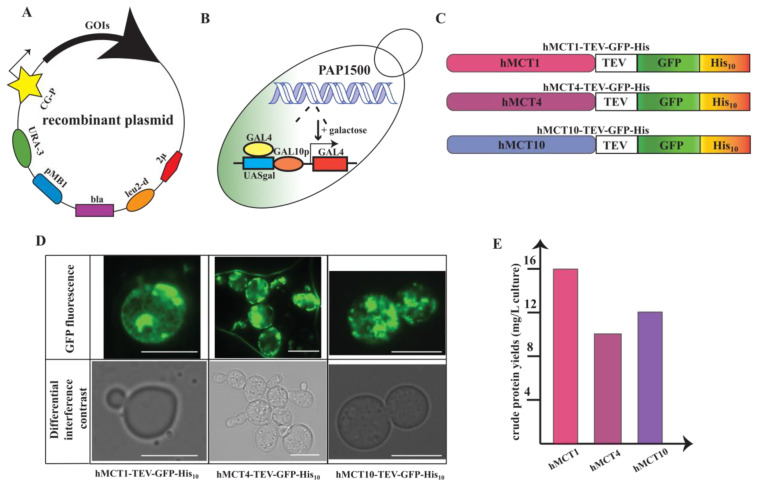
Overview of production of hMCT–TEV–GFP–His_10_ fusions in *S. cerevisiae*. (**A**) Map of the *S. cerevisiae* expression plasmid encoding the different hMCT genes of interest (GOls) fused C-terminally with a tobacco etch virus (TEV), green fluorescent protein (GFP), and a deca-histidine stretch (His_10_). Additional crucial elements in the construct include 2 μ yeast (yeast 2 micron origin of replication), leu2-d (poorly expressed allele of the β-isopropylmalate dehydrogenase gene with truncated promotor), bla (β-lactamase gene), pMB1 (pMB1 an origin of replication), URA3 (yeast orothidine-S-phosphate decarboxylase gene), and CG-P (hybrid promoter of GAL 10 upstream activating sequence and 5’ non-translated leader of cytochrome-1 gene). (**B**) The *S. cerevisiae* protein production strain used in this study is called PAP1500. Upon galactose induction, the PAP1500 overexpresses the Gal4 transcriptional activator. GAL 10 p and UASgal constitute a specific DNA binding site for Gal4 activator. (**C**) outline of the hMCT1–, hMCT4–, and hMCT10–TEV–GFP–His_10_ constructs used in this study. (**D**) Live-cell bioimaging of *S. cerevisiae* cells derived from 2 L cultures after 48 h of induction at 15 °C, the scale bars represent 10 µm. For each construct, GFP fluorescence and differential interference contrast micrographs are shown. (**E**) GFP-based estimates of protein production levels of all constructs used in this study.

**Figure 3 cells-13-01585-f003:**
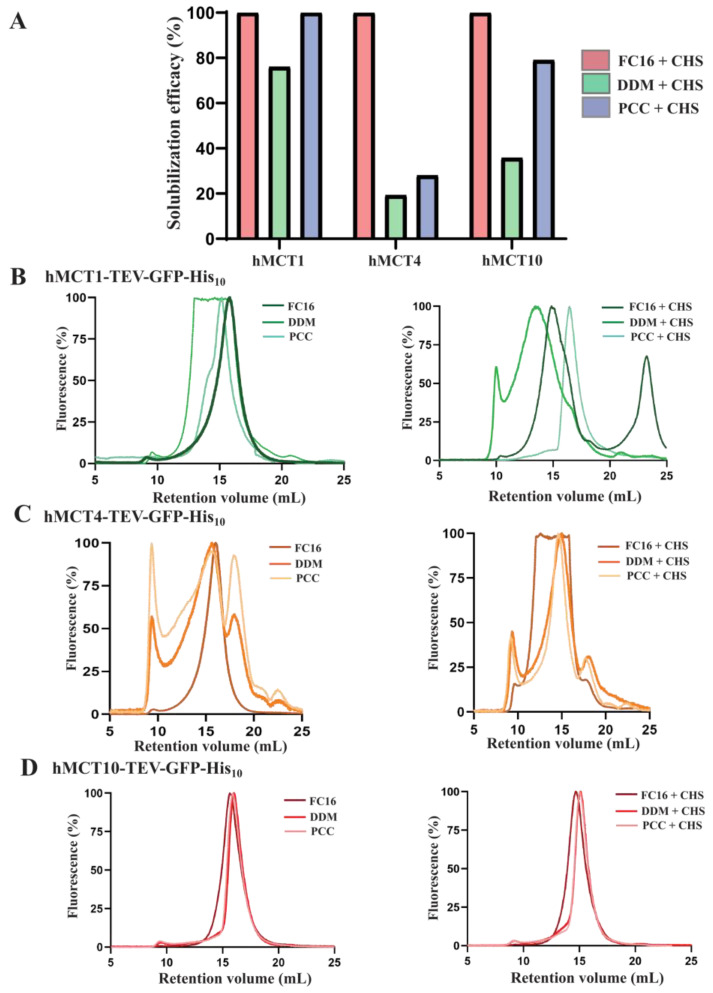
Detergent screening and fluorescent-detection size-exclusion chromatography (F-SEC) analysis of hMCT transporters. (**A**) Solubilization efficiency of hMCT1–, hMCT4–, and hMCT10–TEV–GFP–His_10_ crude membranes isolated from 2 L cultures of *S. cerevisiae* cells. All membranes were solubilized for 2 h at 4 °C in FC-16, DDM, and PCC (final concentration 1%, *w*/*v*) with or without CHS (0.2%, *w*/*v*), and the GFP fluorescence of the supernatant following ultracentrifugation was used to calculate percentage of the solubilization efficiency. (**B**–**D**) F-SEC analysis of detergent-solubilized crude membranes overexpressing hMCT1–, hMCT4–, and hMCT10–TEV–GFP–His_10_, respectively. All membranes were prepared as in (**A**) and the solubilized fraction separated via ultracentrifugation was applied to a Superdex 200 Increase 10/300 GL column where the GFP fluorescence of the eluate was monitored.

**Figure 4 cells-13-01585-f004:**
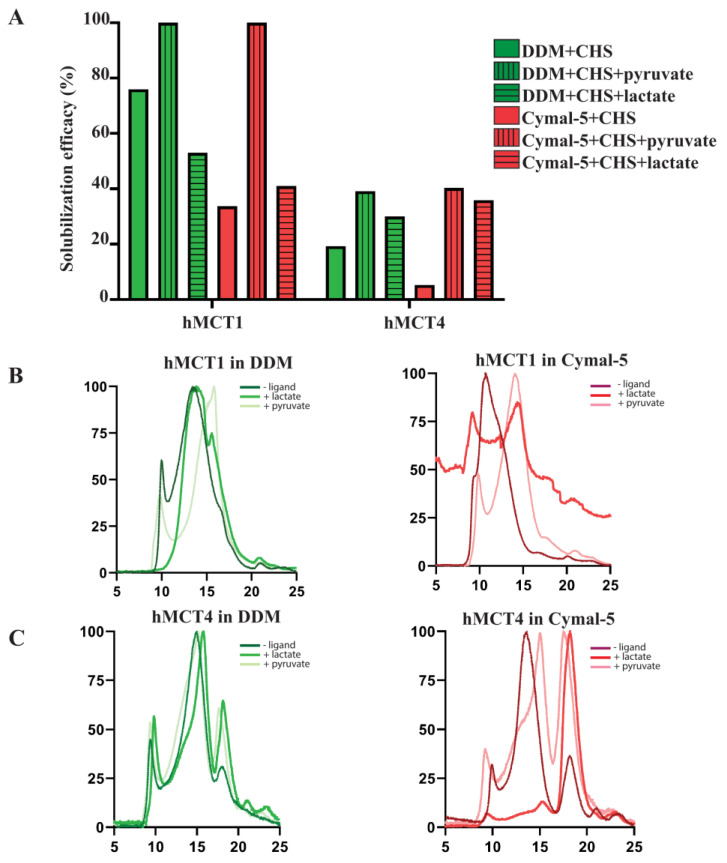
Effect of monocarboxylate ligand presence on solubilization efficiency and protein stability for hMCTs. (**A**) Solubilization efficiency of two main monocarboxylate transporters hMCT1 and hMCT4 in the absence and presence of two of the most abundantly transported ligands pyruvate and lactate. Crude membranes were obtained from 2 L of flask-grown *S. cerevisiae* cells solubilized in the mentioned detergent for 1.5 h at 4 °C, in the presence or absence of 100 mM sodium pyruvate or 100 mM sodium lactate. The GFP fluorescence in the ultracentrifugation supernatant was measured to calculate the solubilization efficiency percentage. (**B**) F-SEC analysis of hMCT1 solubilized in DDM and Cymal-5 in the absence and presence of monocarboxylate ligands. Supernatants from (**A**) were collected and applied on a Superdex 200 Increase 10/300 GE column with continuous monitoring of the GFP signal. (**C**) F-SEC analysis of hMCT4 solubilized in DDM and Cymal-5 in the absence and presence of monocarboxylate ligands.

**Figure 5 cells-13-01585-f005:**
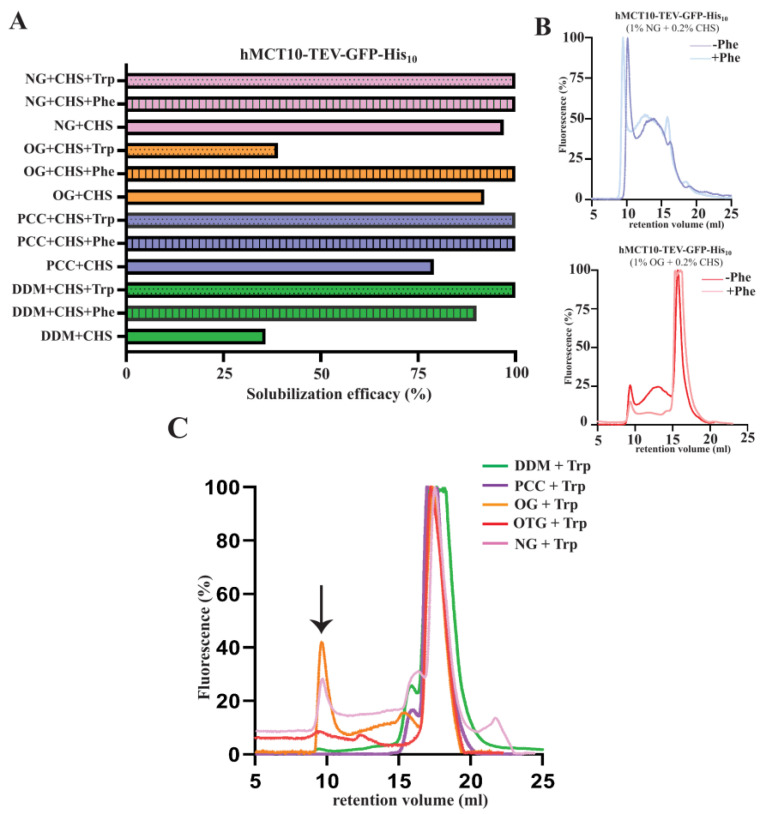
Effect of Phe and Trp on solubilization efficiency and stability of amino acid transporter hMCT10. (**A**) Crude membranes prepared from *S. cerevisiae* cells obtained from a 2 L culture tested for the impact of the presence of aromatic amino acids tryptophan and phenylalanine on the percentage extracted from cell membranes. Each detergent mentioned was used at a final concentration of 1% (*w*/*v*) with 0.2% (*w*/*v*) CHS and, additionally, either 10 mM Trp or 10 mM Phe was added. GFP fluorescence was measured in the solubilized material and used to calculate the percentage solubilized. (**B**) F-SEC analysis of solubilized material (obtained from panel **A**) for the glucoside detergents in the presence and absence of Phe. The solubilized fraction was applied to a Superdex 200 Increase 10/300 GE column where the GFP fluorescence of the eluent was followed and displayed in the chromatogram. (**C**) F-SEC analysis to visualize the effect of the presence of Trp on the stability of hMCT10–TEV–GFP–His_10_, with the black arrow indicating the void peak.

**Figure 6 cells-13-01585-f006:**
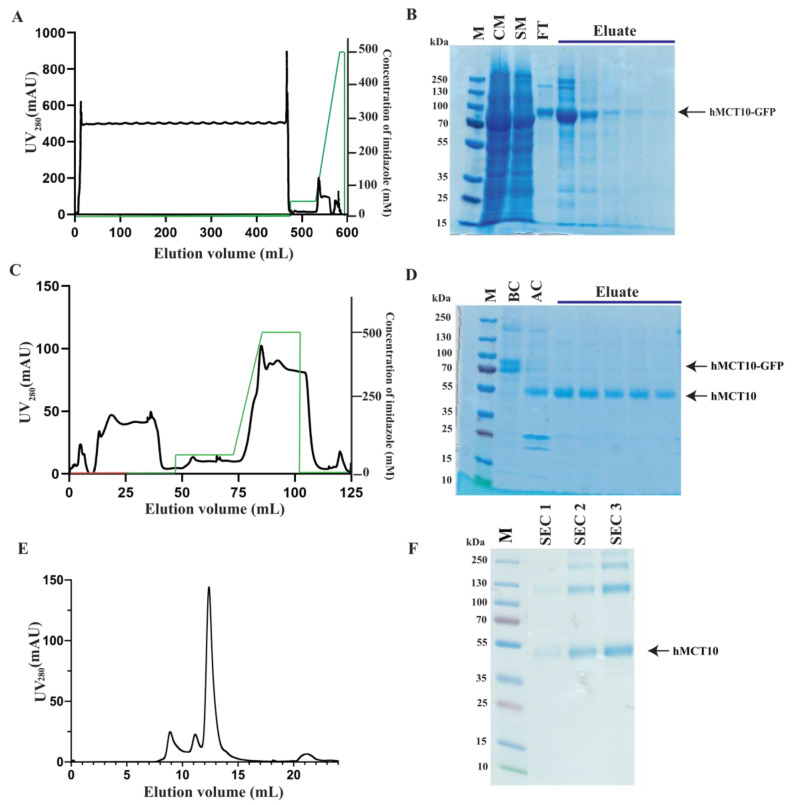
Purification of hMCT10–TEV–GFP–His_10_. (**A**) Representative immobilized metal affinity chromatography (IMAC) chromatogram of hMCT10–TEV–GFP–His_10_. Membranes were prepared from 40 g of *S. cerevisiae* cells grown in fermenter and solubilized for 4 h at 4 °C in PCC (1% *w*/*v*, final concentration) supplemented with CHS (0.2% *w*/*v*, final concentration) and 5 mM Phe. IMAC profile displays UV_280 nm_ signal for eluted protein (black) and the corresponding imidazole gradient used for eluting bound protein (green). (**B**) Coomassie-stained SDS-PAGE analysis of samples collected during IMAC in the presence of PCC. M: marker; CM: crude membranes; SM: solubilized membranes; FT: IMAC flow-through; Eluate: top IMAC fractions collected during IMAC. Arrow indicates gel band corresponding to hMCT10–TEV–GFP–His_10_. (**C**) Representative reverse immobilized metal affinity chromatography (RIMAC) chromatogram of hMCT10–TEV–GFP–His_10_ performed in the presence of PCC. Protein eluate from B was subject to 16 h TEV cleavage in a 1:5 TEV to protein ratio under dialysis. RIMAC profile displays UV_280 nm_ signal for protein (black), sample application phase (red), and the imidazole gradient applied (green). (**D**) Coomassie-stained SDS-page analysis of samples collected during RIMAC in the presence of PCC. M: marker; BC: before cleavage; AC: after cleavage; Eluate: top RIMAC fractions collected during RIMAC. Arrows indicate pre-cleavage hMCT10–GFP and successfully cleaved sample hMCT10. (**E**) Size-exclusion chromatography (SEC) profile in the presence of LMNG of RIMAC-pure-cleaved hMCT10 separated on a Superdex 200 Increase 10/300 GE column with UV_280 nm_ monitored. (**F**) Coomassie-stained SDS-page analysis of SEC pure protein. M: marker, SEC1: SEC fraction 1; SEC2: SEC fraction 2; SEC3: SEC fraction 3. Arrow indicates SEC-pure-cleaved hMCT10, including several oligomeric states.

## Data Availability

Uncropped gels of for the figures included in this work can be downloaded from the Appendix A.

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
