# Peer review of "Isolation of Functional Human MCT Transporters in Saccharomyces cerevisiae"

_cells, 2024, doi:10.3390/cells13181585_

Round 1

Reviewer 1 Report

Comments and Suggestions for Authors

In the present manuscript, the authors describe expression of  Human monocarboxylate transporters (hMCTs) 1,4 and 10 in Saccharomyces cerevisiae, and an efficient procedure for high-scale purification of the MCT10. These proteins are very interesting targets, but difficult to express, isolate and store in soluble form, which poses a problem in their characterization and structural studies conductive to their “druggability”. The choice of the expression system is novel and exciting as the yeast presents a eukaryotic organism with potential advantage in protein folding over E. coli, but is easier and less expensive to handle as insect or mammalian cell systems. The authors first express GFP-fused proteins  and use fluorescent imaging to confirm the expression, and then isolate yeast membranes. Detergent screening is applied to optimize solubilization efficiency, and Fluorescence-detection size-exclusion chromatography is used to monitor the dispersity of the product. The effect of aromatic amino acids Phe and Trp on solubilization efficiency and stability hMCT10-TEV-GFP-His10 is investigated, and it is established that Phe is beneficial for the extraction procedure. Large-scale purification of fermentor-produced hMCT10-TEV-GFP-His8 is then described, and the target protein is indeed isolated as a highly pure species, the only feature to work on is the oligomerization which appears after TEV-cleavage of the product.

The manuscript is very well written, the experiments are well described and the reasoning behind the experimental plot easy to follow. My only question/remark would be if the authors could explain how the significance of the experimental parameters and outcomes was assessed in (i) detergent screen) and (ii) Phe /Trp as additive was evaluated, as only a single value is presented – what is the typical deviation in such experiments?

Please find below a list of minor remarks which I hope you will find helpful.

Line 19: for the readers, it would be important to be more specific about the expression system, for example: whole-cell S. cerevisiae system, or similar

Line 64: citation and source of the figure (PDB ID, modelling program…)should be cited here.

Line 134: S. cerevisiae in italics

Line 168: cell pellets were surely not dissolved in 0.9% NaCl, do you mean washed and resuspended?

Line 179: please specify the diameter of glass beads. “8 rounds of 2 minutes each“ – was there a machine used? Please be more precise.

Line 217: as previously published (25).

Line 227: through centrifugation

Line 235: “32000 rpm for 1 hour at 4 °C.” – please g units

Line 300: 2u, micron sign instead

Figure 2D: a negative control (such as uninduced cells, or similar) should be added

Author Response

See the attached word-file.

Reviewer 2 Report

Comments and Suggestions for Authors

The article entitled "Isolation of functional human MCT transporters in Saccharomyces cerevisiae” aims to develop a straightforward, cost-effective strategy for the overproduction of hMCTs using a Saccharomyces cerevisiae-based system.

Can you explain why you chose Saccharomyces cerevisiae as the expression system rather than more common systems such as E. coli or mammalian cells? Have comparative studies been carried out to justify this choice?

Given the known problems with protein stability in yeast systems, how do you ensure that the expressed hMCTs are functionally equivalent to those in human cells?

Your study indicates a high protein yield, but could you explain how you assessed the functional integrity of these proteins after purification?

How do you explain the different localization (plasma membrane vs. intracellular compartments) of the expressed proteins and how might this affect their function?

Were negative controls used to confirm the specificity of the detergent screening results? If so, what were the results?

How did you ensure that the observed effects of ligands such as pyruvate and lactate on solubilization efficiency were not due to indirect factors such as changes in buffer conditions?

You mention that hMCT10 was successfully purified with considerable yields. How does this result translate to potential therapeutic applications or drug development?

Given the structural differences between human MCTs and their bacterial homologs, how do you ensure that the findings from this yeast-based system are relevant to human physiology?

What are the next steps to confirm that the purified proteins retain their structural and functional properties over time?

How do you plan to address the potential limitations of the yeast expression system in future studies?

Author Response

See the attached word-file.

Round 2

Reviewer 2 Report

Comments and Suggestions for Authors

The authors have responded to my concerns.